# An open-source interactive AI framework for assisting automatic literature review in forensic medicine: Focus on brain injury mechanisms

**Ya-Wen Liu[1,2], Dong-Hua Zou[1], He-Wen Dong[1], Yuan-Yuan Liu[1,3], En-Hao Fu[1,3], Zhi-Ling Tian[1]\*, Ning-Guo Liu[1]\***

**1** Academy of Forensic Science, Shanghai Key Laboratory of Forensics Medicine, Shanghai Forensic Service Platform, Key Laboratory of Forensic Medicine, Ministry of Justice, Shanghai, People's Republic of China, **2** School of Forensic Medicine, Shanxi Medical University, Jinzhong, Shanxi, People's Republic of China, **3** Henan University of Science and Technology, School of Basic Medicine and Forensic Medicine, Institute of Medical Aspects of Specific Environments, Forensic Medicine Identification Center, Luoyang, Henan, People's Republic of China

\* Tianzl@ssfjd.cn (ZLT); Liung@ssfjd.cn (NGL)

## Abstract

### Background and Objective

Systematic reviews and meta-analyses are critical in forensic medicine; however, these processes are labor-intensive and time-consuming. ASReview, an open-source machine learning framework, has demonstrated potential to improve the efficiency and transparency of systematic reviews in other disciplines. Nevertheless, its applicability to forensic medicine remains unexplored. This study evaluates the utility of ASReview for forensic medical literature review.

### Methods

A three-stage experimental design was implemented. First, stratified five-fold cross-validation was conducted to assess ASReview's compatibility with forensic medical literature. Second, incremental learning and sampling methods were employed to analyze the model's performance on imbalanced datasets and the effect of training set size on predictive accuracy. Third, gold standard were translated into computational languages to evaluate ASReview's capacity to address real-world systematic review objectives.

### Results

ASReview exhibited robust viability for screening forensic medical literature. The tool efficiently prioritized relevant studies while excluding irrelevant records, thereby improving review productivity. Model performance remained stable when labeled training data constituted less than 80% of the total sample size. Notably, when the training set proportion ranged from 10% to 55%, ASReview's predictions aligned closely with human reviewer decisions.

**Data availability statement:** All data underlying the findings described in our manuscript are freely accessible to other researchers. The corresponding data are available within the paper itself.

**Funding:** This study was financially supported by the following projects: The Central Research Institute Public Project (GY2024D-1, GY2024Z-1) provided the computational equipment for this study.The Shanghai Key Laboratory of Forensic Medicine (21DZ2270800), the Shanghai Forensic Service Platform, the Key Laboratory of Forensic Science (Ministry of Justice), and the Project of Shanghai Association of Forensic Science (SHSFJD2023-008) provided the laboratory space and configured the computational environment for the model.The Shanghai Yangfan Special Programme (23YF1448700) provided forensic pathologists with the required experimental qualifications.

**Competing interests:** The authors have declared that no competing interests exist.

## Conclusion

ASReview represents a promising tool for forensic medical literature review. Its ability to handle imbalanced datasets and gather goal-oriented information enhances the efficiency and transparency of systematic reviews and meta-analyses in forensic medicine. Further research is required to optimize implementation strategies and validate its utility across diverse forensic medical contexts.

---

## Introduction

The rapid proliferation of scientific literature across disciplines has significantly complicated the task of reviewing and filtering relevant studies, requiring meticulous evaluation by researchers [1–2]. In forensic pathology, determining the cause of death involves integrating knowledge from a broad array of scientific fields. Given the high volume of cases, forensic experts often lack the time to thoroughly review extensive literature. However, ensuring the accuracy and fairness of forensic evidence in court demands rigorous support from the literature. Forensic pathologists must therefore not only delve into their own field but also explore research findings from related disciplines. This comprehensive approach is essential to meet the complex demands of case analysis. However, the challenge lies in eliminating irrelevant information and extracting necessary data from a vast pool of literature, underscoring the need for sophisticated data filtering methodologies.

Advancements in artificial intelligence (AI) and machine learning offer powerful tools for systematic literature review and validation [3–5]. Sophisticated machine learning algorithms can swiftly identify and critically assess relevant publications, thereby enhancing research integrity and efficiency [6]. However, several challenges remain unresolved:

1. The adaptability of AI tools, developed in disparate fields, to forensic medical remains uncertain.

2. AI models must accurately discern and assimilate relevant information from voluminous datasets while filtering out superfluous data.

3. Models need to identify innovative methodologies across seemingly unrelated disciplines through analogical reasoning.

4. The usability of AI tools must be simplified to accommodate forensic professionals with varying levels of computational expertise.

The application of open-source machine learning frameworks in forensic medical literature review remains largely unexplored. A notable exception is ASReview, an interactive framework introduced by Rens van de Schoot et al. [7], which enhances the efficiency and transparency of systematic reviews through active learning and automated literature categorization. While ASReview has shown promise in other domains, its performance in forensic pathology remains unvalidated.

To address these gaps, we proposed a three-phase study using ASReview to evaluate its effectiveness in forensic medical literature review, using deep learning in traumatic brain injury research as a test case. This field, characterized by its interdisciplinary nature and large volume of literature, presents significant challenges for accurate literature retrieval. Our study aimed to:

1. Assess the feasibility of using ASReview for forensic literature reviews through stratified five-fold cross-validation.

2. Evaluate its performance with imbalanced datasets by manipulating relevance ratios and dataset sizes.

3. Investigate the model's decision-making processes.

4. Evaluate the usability of ASReview for non-computer science professionals.

Our goal was to introduce an efficient automated method for forensic medical literature review, identify optimal parameter settings, and explore the potential of machine learning models in this domain.

## Data preparation and model configuration

### Data collection and filtering

Based on our research topic of "the application of deep learning in traumatic brain injury research," we established search criteria to retrieve primary literature. Considering the characteristics of the forensic medicine field, PubMed was selected as the search platform. Due to the rapid increase in the number of scholarly articles pertaining to the application of deep learning in medical image analysis since 2015 [8–9], we queried PubMed [10] for papers containing "deep learning and brain injury" or "deep learning and head injury" in the title, keywords, or abstract from January 1, 2014 to March 5, 2024 [11]. We chose the "most recent" strategy, which meant retrieving articles in reverse chronological order of publication. This approach helped maintain data randomness. The "best match" option in PubMed uses algorithms to assign article weights, which may introduce bias. Similarly, criteria like author name or nationality could create bias. A total of 419 articles were retrieved using this strategy. We downloaded all the articles (avoiding manual selection) and imported them into Zotero 6.0.37 in the original chronological order from the search. After importation, we did not use Zotero's sort and screen options. This approach avoided the influence of Zotero's algorithms or programs on the data, thereby maximizing data randomization.

Then we eliminated problematic papers. Before the data cleaning process, we established the following criteria for "problematic papers": 1. Retracted articles: These included articles retracted by authors and editors, which could be identified by retraction marks or statements on PubMed. 2. Special-format documents: Only literature from journal papers, conferences, and books was reserved. 3. Duplicated publications. 4. Duplicated collections: As two keyword combinations were used for retrieval, some articles might have been retrieved twice. For such cases, only one copy of the article was retained.

Ultimately, a total of 329 references were identified as valid. Each paper was assigned a unique index number in the order of collection and filtering, which is referred to as the "index order" throughout this study. This index order was used to systematically organize and reference the dataset during the experimental process.

### Gold standard

For our experimental design, we prepared two distinct truth sets using the same underlying dataset but with different classification criteria. Truth 1 was utilized to investigate the feasibility of applying the model in forensic science and to explore the optimal parameter range for its use. Truth 2 was developed to compare the discrepancies between machine learning results and human searchers' expectations, as well as to investigate the decision-making processes of the machine model.

For Truth 1, three senior forensic experts with extensive knowledge in both craniocerebral injury and deep learning independently reviewed the titles and abstracts of each paper and performed a binary classification (relevant-irrelevant) according to established criteria. In this study, craniocerebral injuries were defined as primary injuries to the skull and brain directly caused by external forces (e.g., breaking, tearing, and stretching of tissue, paralysis of axonal transport, and functional effects of percussion) or as indirect consequences of primary injuries (e.g., hemorrhage, ischemic injury, inflammation, swelling) [12]. Pathological changes caused by diseases or physiological factors (e.g., aging, mental state) were excluded. Additionally, the scope of deep learning techniques was confined to the analysis of cranial medical imaging and the study of traumatic brain injury (TBI) patterns. When an article meets the above criteria, it is deemed relevant; otherwise, it is considered irrelevant. When two or more experts agreed on the relevance of a paper, it was marked as relevant and encoded as 1; otherwise, it was marked as irrelevant and encoded as 0. This binary classification simulated computer language, facilitating straightforward comparison between the model's output and human expectations. Based on this criterion, the 329 valid pieces of literature were stratified into 70 relevant and 259 irrelevant works, forming Truth 1 for the machine learning review.

For Truth 2, the same three experts categorized the 70 relevant papers into five levels of relevance using a ranking system ranging from 1 (most relevant) to 5 (least relevant) (Table 1). These papers were then sorted in descending order of relevance. The 259 irrelevant papers retained their coding from Truth 1 (encoded as 0). Thus, a sequence of codes represented by 0, 1, 2, 3, 4 and 5 was obtained to indicate varying degrees of article relevance. This sequence constitutes Truth 2. This grading system aimed to simulate real-world search expectations, investigating whether the model could handle various levels of relevance, from "fully meeting requirements" to "partially meeting requirements" and "not directly relevant but enlightening," while excluding a large amount of irrelevant content.

In the gold standard, there were two assessments of expert-rating reliability. First, when the 329 articles were divided into relevant and irrelevant categories, a Fleiss' kappa value of 0.85 was obtained. Second, for the five-level rating of the 70 relevant articles, the Fleiss' kappa value was 0.83. These results indicated good consistency among the experts' ratings, enabling their use as a gold standard. Notably, we required that a rating label be agreed upon by at least two experts to be valid. This approach further reduced potential discrepancies among the experts and improved the consistency of the labels.

## Model configuration

The Active Learning for Systematic Reviews toolbox (van de Schoot et al., 2021) Version 1.6

The model features a simple yet extensible default configuration, which includes a naive Bayes classifier, TF-IDF feature extraction, dynamic resampling for balancing strategy, and a query strategy based on deterministic sampling (Table 2). The selection of these default values is justified by their consistently high performance across several benchmark experiments on various datasets. For example, the Naive Bayes (NB) classifier has been widely proven to perform well in text classification tasks, especially in systematic review screening [13–14]. Additionally, the short computation time of these default settings makes them appealing for applications, as the software should be capable of running locally. Specifically, the combination of NB and TF-IDF demonstrated a relatively short average computation time across all datasets, typically ranging from a few seconds to tens of seconds, depending on the size and complexity of the dataset [15–17]. This short computation time renders these default settings more efficient in practical applications, enabling rapid responses to reviewers' decision-making needs.

Furthermore, users can incorporate their own classifiers, feature extraction techniques, query strategies, and balancing strategies.

## External validation

We collected entirely new data using the same procedures as those for the training set to conduct external validation of the model's performance. The training set covers from January 1, 2014 to March 5, 2024, while the external validation set

**Table 1. Expert grading criteria at five levels.**

| Classifier | Criterion | Level |
|---|---|---|
| Level 1 | Develop a medical image analysis model for cranial trauma utilizing deep learning technology, ensuring a clear analysis and validation of the classification or mechanisms of specific types of injuries,<br>OR<br>Conduct a comprehensive overview of the field where deep learning is applied to the etiology of cranial injuries, highlighting the deep learning models and algorithms utilized in the medical imaging analysis of such injuries, | 1 |
| Level 2 | Address the indeterminate aspects that fall between Level 1 and Level 3, where the exact classification or relevance cannot be conclusively determined, | 2 |
| Level 3 | A deep learning model has been proposed, yet it lacks a definitive analysis and validation of the classification or mechanisms pertaining to specific types of injuries,<br>OR<br>Employ deep learning technology to develop a biomechanical or finite element model for reconstructing the process of cranial injury, and explore the mechanisms and etiology of the injury, | 3 |
| Level 4 | Address the indeterminate aspects that fall between Level 3 and Level 5, where the exact classification or relevance cannot be conclusively determined, | 4 |
| Level 5 | Improvements in imaging technology are discussed,<br>OR<br>Advancements in the field of computer science have led to significant improvements in algorithms,<br>OR<br>Clinical medicine has increasingly incorporated artificial intelligence (AI) to enhance various aspects of patient care, including auxiliary diagnosis, prognosis prediction, and the determination of surgical and therapeutic indications,<br>OR<br>Others. | 5 |

The rating criteria in the Table 1 simulate the actual retrieval requirements of the users. Three experts independently conducted a double-blind experiment to ensure the fairness and rationality of the scoring results. The ranking system ranges from 1 to 5, with '1' indicating the highest relevance and '5' the lowest.

includes all literature from March 6, 2024 to May 25, 2025, totaling 116 articles. After data cleaning, 109 valid articles were obtained.

## Statistical analysis tools

IBM Corp. IBM SPSS Statistics for Mac [version 26.0.0.0]. Armonk, NY: IBM Corp.; 2020.

## Ethical conduct

This article does not contain any studies involving direct research on human participants or animal experiments conducted by the authors, as the research was conducted using deep learning models and computer simulation methods.The materials utilized in this experiment are exclusively comprised of academic papers that have been published in the public domain.Therefore, ethical review is not required.

## Informed consent

In this study, the human participants involved were limited to three experts who classified the relevance of the papers, all of whom provided informed consent regarding the content of the experiment.

## Methods

Our experimental design was divided into three phases (Fig 1). Initially, we implemented stratified five-fold cross-validation to preliminarily evaluate the applicability of ASReview to forensic literature. If the results were positive, indicating that ASReview was suitable for the field of forensic science, we proceeded with the second and third

**Table 2. Selection and function of model parameters.**

| Functional module | Parameter | Function |
|---|---|---|
| Feature extraction technique | TF-IDF | To identify and extract relevant information from data and to convert text data into a format understandable by the models. |
| Classifier | naive bayes | Classifiers are utilized to assign data items to one or more categories. The objective of a classifier is to predict the class labels of the data based on its features. |
| Query strategy | maximum | It refers to a methodology or technique employed for the efficient retrieval or extraction of information from voluminous datasets. |
| Balance strategy | dynamic resampling (double) | To handle data imbalance. |

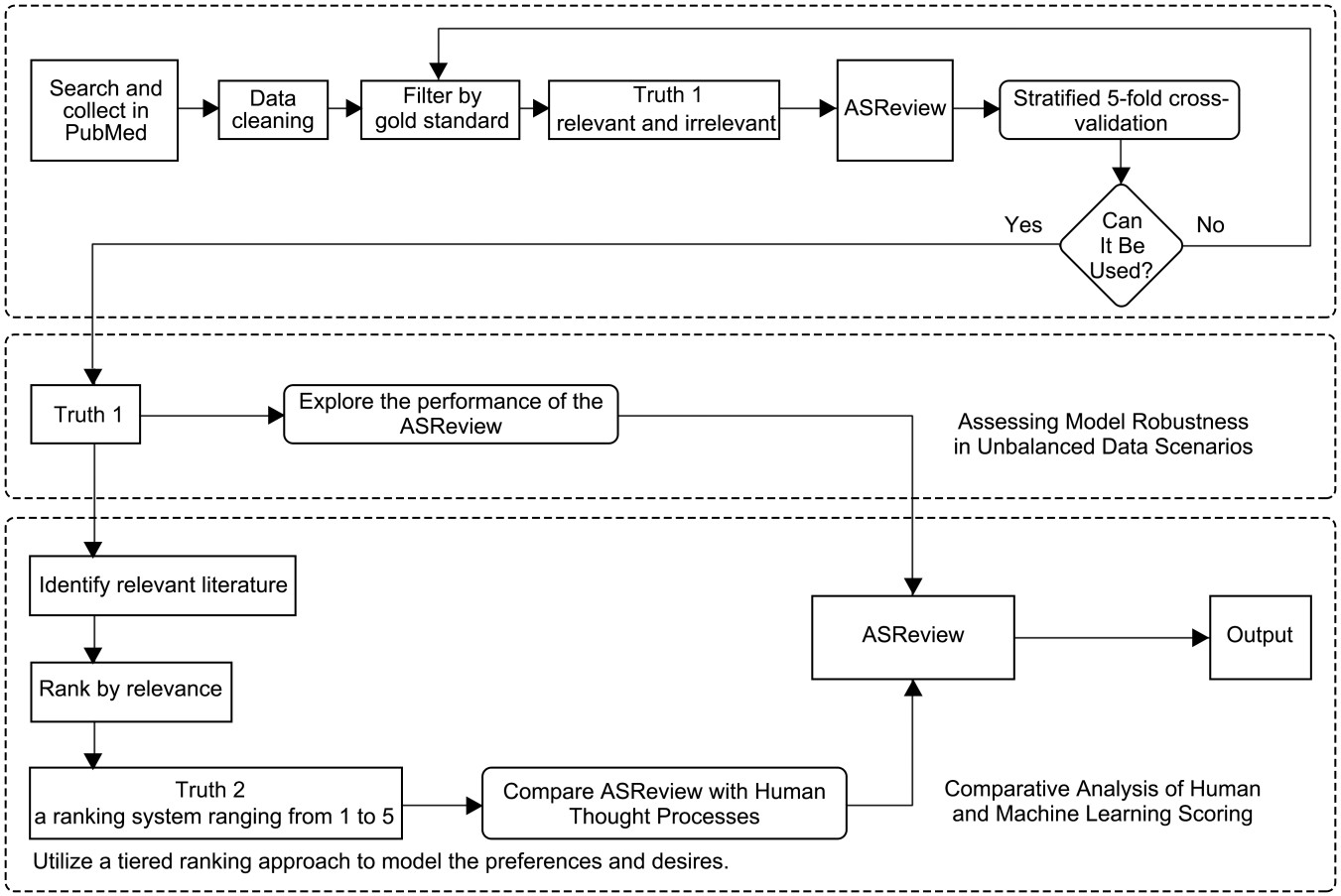

**Fig 1. Experimental Procedure Schematic of three experiments The diagram represents the three components of the experiment, namely Stratified 5-fold Cross-Validation, Assessing Model Robustness in Unbalanced Data Scenarios, and Comparative Analysis of Human and Machine Learning Scoring.** The outcome of the Stratified 5-fold Cross-Validation serves as a prerequisite for the progression to subsequent experimental phases. If the results are positive, suggesting that ASReview is apt for the forensic science domain, we advance to the second and third stages of the experiment; if the results are negative, indicating a lack of suitability, we return to the processes of data collection and cleansing.

experimental phases; if the results were negative, we would have reverted to the data collection and cleaning process. Subsequently, we employed an incremental learning sampling technique to investigate the model's performance on imbalanced datasets and the impact of training set size on its accuracy. Finally, we evaluated ASReview's capacity to

meet real-world review expectations by translating reviewer requirements into computational language. In addition, we selected entirely new data using the same procedures as those for the training set to conduct external validation of the model's performance.

In this study, the human participants involved were limited to three experts who classified the relevance of the papers, all of whom provided informed consent regarding the content of the experiment. The duration of the experiment spanned from February 20, 2024, to November 20, 2024.

## Stratified 5-fold cross-validation

In order to assess the model's applicability within the realm of forensic pathology, stratified five-fold cross-validation was employed to uphold a consistent ratio of relevant and irrelevant literature. Given the limited dataset size, five-fold cross-validation was selected over a ten-fold approach. Under the Oracle Interactive AI framework, stratified five-fold cross-validation was conducted according to the model's default settings. The specific process is as follows:

a. The current subset was designated as the validation set.

b. The remaining four subsets were combined to form the training set.

c. The ASReview model was trained using the compiled data from the training set.

d. The model's performance was evaluated using the validation set.

This process was iterated until each subset had served as the validation set (Fig 2). During each iteration, performance metrics such as accuracy, precision, recall, and the F1 score were recorded. Additionally, specificity was analyzed [18]. The average values of these metrics across all iterations were then calculated to provide an overall assessment of the model's performance.

## Assessing model robustness in unbalanced data scenarios

The purpose of this experiment was to explore the performance of the model on imbalanced datasets and to examine the impact of training set size on this performance. The term "imbalance" in this context refers to the significant predominance of irrelevant data over relevant data within the dataset. This was intentionally designed to explore the model's capability to

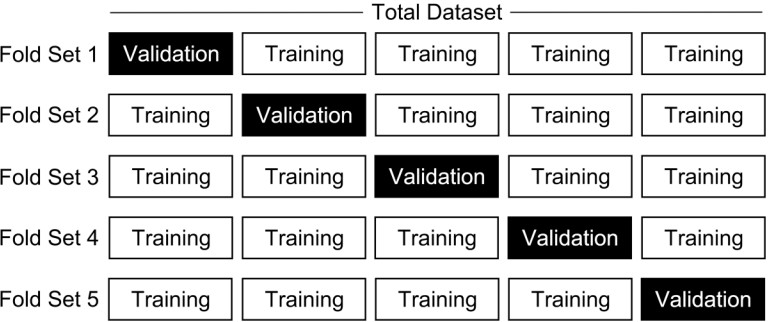

**Fig 2. Schematic Representation of Stratified Five-Fold Cross-Validation Process.** The specific process is as follows: a. The current subset was designated as the validation set. b. The remaining four subsets were combined to form the training set. c. The ASReview model was trained using the compiled data from the training set. d. The model's performance was evaluated using the validation set. This process was iterated until each subset had served as the validation set.

accurately identify target information and filter out a substantial amount of redundant content. Based on this, we aimed to further determine the performance range of the model during its application.

All entries from the Zotero library were exported and subsequently converted into CSV (Comma-Separated Values) format for further data analysis.

This section of the experiment was divided into two parts. Initially, following the default settings of ASReview, the first five relevant and the first five irrelevant articles from Truth 1 were extracted to serve as the training set. The remaining articles constituted the test set. The training and test subsets were utilized to establish the control group for comparative analysis. Subsequently, we divided Truth 1 into two subsets: a relevant group (70 articles) and an irrelevant group (259 articles). Through stratified sampling, we extracted the top 5% of articles from each subset to form a training set, ensuring that the training set was representative in terms of relevance and irrelevance. The articles not selected from each subset automatically formed the test set, which was used to validate the model's training performance. We iterated and repeated this sampling process, increasing the size of the training set by 5% each time, until we obtained a total of 18 different combinations of training and test sets of varying sizes. This process allowed us to assess the impact of different training set sizes on model performance (Fig 3, Table 3).

Due to the limited sample size and the utilization of the complete ASReview machine learning framework for training without the need for parameter tuning, a validation set was not established. Additionally, extracting the training sets in accordance with the index order ensured that each new training set completely included the content of the previous one. This meant that each subsequent machine learning iteration was built upon the previous one, rather than initiating new learning sessions. This incremental learning approach effectively controlled the variables and maintained consistency throughout the learning process.

The control group and the 18 training sets were subsequently subjected to the model for analysis, yielding the final outcomes. The review's outcomes will subsequently be exported into an Excel-compatible file format, enabling thorough data processing and subsequent statistical analysis to ensure the derivation of robust and valid conclusions. The entire process described above was repeated once.

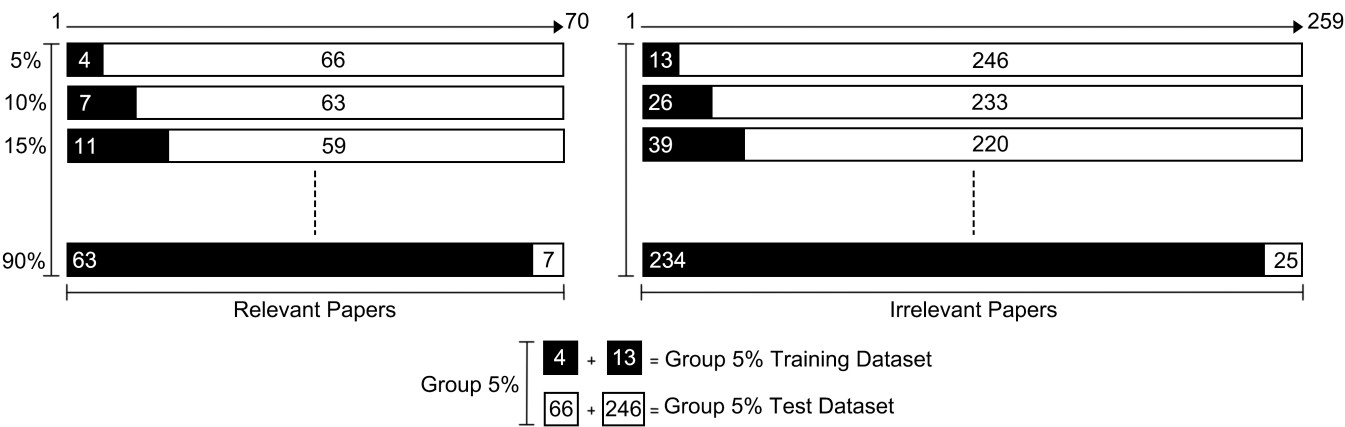

**Fig 3. Schematic Diagram of Stratified Sampling.** The dataset is stratified into groups of relevant and irrelevant papers for stratified sampling. Within each group, the papers are sequentially numbered, and a 5% increment of the total number is sampled in each round. The samples from both groups are then pooled to constitute the test set. For instance, in the first round, papers numbered 1 to 4 from the relevant group and 1 to 13 from the irrelevant group are selected. This process is repeated, ensuring that each subsequent test set includes the content of the previous one. The sampling method maintains a consistent ratio of relevant to irrelevant papers throughout, ensuring that the learning process is cumulative and builds upon prior knowledge rather than starting anew.

**Table 3. Stratified sampling data distribution.**

| | Training dataset | | | Test dataset |
|---|---|---|---|---|
| | **Relevant** | **Irrelevant** | **Total** | |
| Default | 5 | 5 | 10 | 319 |
| 5% | 4 | 13 | 17 | 312 |
| 10% | 7 | 26 | 33 | 296 |
| 15% | 11 | 39 | 50 | 279 |
| 20% | 14 | 52 | 66 | 263 |
| 25% | 18 | 65 | 83 | 246 |
| 30% | 21 | 78 | 99 | 230 |
| 35% | 25 | 91 | 116 | 213 |
| 40% | 28 | 104 | 132 | 197 |
| 45% | 32 | 117 | 149 | 180 |
| 50% | 35 | 130 | 165 | 164 |
| 55% | 39 | 143 | 182 | 147 |
| 60% | 42 | 156 | 198 | 131 |
| 65% | 46 | 169 | 215 | 114 |
| 70% | 49 | 182 | 231 | 98 |
| 75% | 53 | 194 | 247 | 78 |
| 80% | 56 | 208 | 264 | 65 |
| 85% | 60 | 221 | 281 | 48 |
| 90% | 63 | 234 | 297 | 32 |

"Default" represents the default settings of ASReview as a control.

"X%"refers to the percentage of the total dataset that is sampled to form the training set.

## Comparative analysis of human and machine learning scoring

The objective of this experimental segment was to elucidate the underlying mechanisms of deep learning, examine the discrepancies between machine learning outcomes and manual analysis, and assess whether the machine-generated results meet the expected goals in real-world scenarios. We continued to use the incremental learning method described in the section "Methods—Assessing Model Robustness in Unbalanced Data Scenarios" to process Truth 2, including a control group (default settings), and obtained 19 combinations of training and test sets. Notably, the "relevant" subset in Truth 2 was derived from a five-level scoring system applied by three experts based on the gold standard criteria. The expert scoring criteria are detailed in the section "Data Preparation and Model Configuration—Gold Standard," with Table 1 provided for reference. This scoring system was designed to simulate a real-world retrieval process. Given the retrieval target of "deep learning" and "mechanisms of traumatic brain injury," which encompasses multiple core concepts and keywords, the following levels were established:

 Level 1: Articles that fully meet all core concepts.

 Level 3: Articles that meet some core concepts.

 Level 5: Articles that do not directly meet the core concepts but have valuable experimental methods or ideas worth referring to.

 Levels 2 and 4: Introduced to address cases where experts were uncertain and found it difficult to classify articles between two levels.

 This approach was designed to evaluate the similarity and difference between machine learning results and human thinking under varying data volumes. Each of the 19 datasets (including the control group) was independently trained to

generate learning results using ASReview. The outcomes were then compared and correlated with the manually scored results from Truth 2 to assess their consistency and accuracy.

## Results

### Stratified 5-fold cross-validation

In our study, since the output data did not exhibit a normal distribution, we employed the bootstrap method to estimate the descriptive statistics and 95% confidence intervals for key variables. The analysis of each variable was based on 5 observations (n = 5), and we conducted 1000 bootstrap resamples to enhance the robustness of our statistical inferences. We reported the accuracy, precision, recall, F1 score, specificity, mean, standard deviation, and 95% confidence interval for each variable (Table 4).

From the analysis results, ASReview demonstrated high performance with an average accuracy of 0.78, an average recall of 0.94, and an average specificity of 0.73, with standard errors of 0.04, 0.09, and 0.07, respectively. However, the results in terms of precision and F1 score were somewhat unremarkable. It is evident that ASReview exhibited a high overall accuracy across all categories in its predictions. Precision and specificity are metrics that reflect the model's performance in predicting positive and negative samples, respectively. Lower precision and higher specificity indicate that the model has a stronger performance in predicting negative samples.

As shown in Table 4, the confidence interval widths for accuracy, precision, recall, F1 score, and specificity under the 95% confidence level were 0.06, 0.007, 0.13, 0.05, and 0.11, respectively. The narrow widths of these intervals indicate the precision of our estimates of ASReview's performance. Additionally, the ROC curve analysis (Fig 4) further demonstrates the high performance of ASReview. In stratified five-fold cross-validation, the AUC of the mean ROC curve reached 0.86, indicating that ASReview has a strong ability to predict the relevance of forensic articles.

As depicted in Fig 5, the darkest shades within the confusion matrices consistently highlighted the model's notable ability to correctly identify negative samples. The model demonstrated optimal performance in accurately identifying negative instances, denoted as True Negatives (TN). This proficiency was a critical measure of the model's overall effectiveness, especially in scenarios where the task was to exclude large volumes of irrelevant literature. This capability was essential for ensuring that the model could efficiently filter out non-pertinent information, thereby facilitating a more focused and streamlined review process.

Overall, the results presented in Table 4 and Fig 5 were largely consistent. In the context of real-world search scenarios, when the model could accurately identify the majority of irrelevant literature, it facilitated human reviewers in more efficiently excluding them. After all, human reviewers may also encounter uncertainties regarding some documents, and

**Table 4. Five-fold cross-validation results.**

| Fold set | Accuracy | Precision | Recall | F1 score | Specificity |
|----------|----------|-----------|--------|----------|-------------|
| 1 | 0.80 | 0.52 | 0.93 | 0.67 | 0.77 |
| 2 | 0.80 | 0.52 | 1.00 | 0.68 | 0.75 |
| 3 | 0.77 | 0.48 | 1.00 | 0.65 | 0.71 |
| 4 | 0.80 | 0.52 | 0.79 | 0.63 | 0.81 |
| 5 | 0.71 | 0.42 | 1.00 | 0.60 | 0.63 |
| Average | 0.78 ± 0.04 | 0.49 ± 0.04 | 0.94 ± 0.09 | 0.65 ± 0.03 | 0.73 ± 0.07 |
| CI (95%) | (0.80,0.74) | (0.52,0.45) | (1.00,0.87) | (0.67,0.62) | (0.78,0.67) |

The numbers 1–5 represent the five iterations of the stratified five-fold cross-validation, with each iteration corresponding to a distinct round of model training and evaluation.

Receiver Operating Characteristic (ROC) - Stratified 5-Fold Cross Validation

**Fig 4. Receiver Operating Characteristic Curve(ROC) – Stratified Five-Fold cross-Validation.**

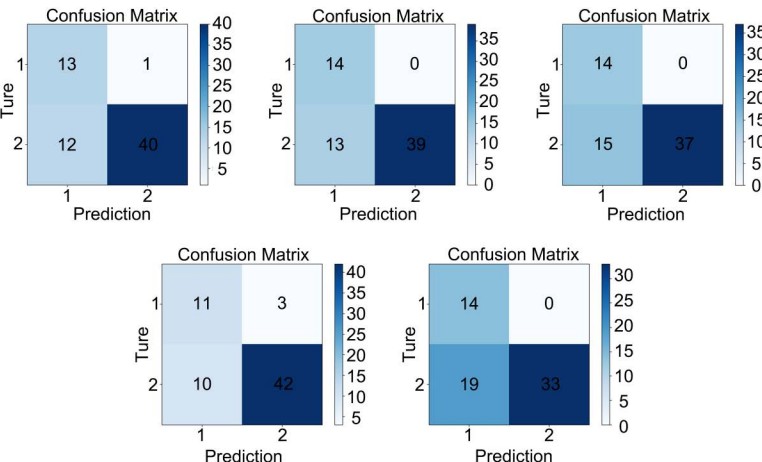

**Fig 5. Stratified Five-Fold cross-Validation Results with a Confusion Matrix.** In Fig 4, the four components of a confusion matrix can be described as follows, with reference to their positions in a clockwise manner from the top left: True Positives (TP): The number of samples that the model correctly predicted as belonging to the positive class. False Positives (FP): The number of negative class samples that the model incorrectly predicted as belonging to the positive class. True Negatives (TN): The number of negative class samples that the model correctly predicted as negative. False Negatives (FN): The number of positive class samples that the model incorrectly predicted as belonging to the negative class. The color intensity within each segment of the confusion matrix represents the predictive performance of the model in that particular direction. A darker shade indicates better predictive performance, while a lighter shade suggests poorer performance. This visual representation allows for a quick assessment of the model's accuracy and the extent of misclassification errors in both the positive and negative classes.

ASReview could retain these for further review. Therefore, it could be concluded that ASReview was capable of handling literature review tasks in the field of forensic science, with a particular strength in filtering out a large amount of irrelevant literature.

## Assessing model robustness in unbalanced data scenarios

To distinguish between the results of two experiments, "Accuracy" is denoted as "a1" and "a2," where "a1" represents the results from the first experiment, and "a2" represents those from the second. Other parameters follow the same naming convention (Table 5). The results presented in Table 5 demonstrate high accuracy, recall, and specificity, while precision and F1 score are relatively low. These findings are consistent with the outcomes of the stratified five-fold cross-validation.

In the present study, we employed Kendall's tau coefficient to assess the ordinal association between two categorical variables: the size of the training dataset and the parameters. Our findings revealed that the mean Kendall's tau coefficient for accuracy was $-0.81$ ($p < 0.01$), for precision was $-0.79$ ($p < 0.01$), and for the F1 score was $-0.80$ ($p < 0.01$), indicating a strong negative correlation between the size of the training dataset and model performance. This result supports the hypothesis that as the training dataset increases, the accuracy, precision, and F1 score tend to decrease. It is worth noting that Kendall's tau coefficient for mean specificity was $-0.66$ ($p < 0.01$), indicating a moderate negative correlation, suggesting that ASReview is more stable and robust in filtering out negative samples. The degree of correlation that recall has with the dataset size remains unknown.

As shown in Fig 6, overfitting was observed when the training set size exceeded 264 samples, which constituted more than 80% of the total samples [19]. The results and average values from the two experiments indicated that the learning curves for accuracy and specificity were relatively concentrated. In contrast, the curves for precision, recall, and F1 score each displayed outlier points. This suggests that ASReview exhibited strong performance in terms of both accuracy and specificity, with a stable and reliable outcome.

In this segment of the experiment, we controlled the size of the dataset and the ratio of relevant to irrelevant data to assess the model's capability to handle various retrieval scenarios. The results indicated that ASReview possessed the ability to address imbalanced data scenarios. Concurrently, overfitting was observed when the training set size exceeded 80% of the total sample, implying that the model's performance began to be constrained.

**Table 5. Model outputs and kendall correlation analysis results from two experiments.**

|  |  | Average | SD | Tau/τ | Sig. |
|---|---|---|---|---|---|
| Accuracy | a1 | 0.80 | 0.07 | −0.86** | 0.00 |
|  | a2 | 0.80 | 0.07 | −0.75** | 0.00 |
|  |  | 0.80 | 0.07 | −0.81 | 0.00 |
| Precision | p1 | 0.54 | 0.09 | −0.82** | 0.00 |
|  | p2 | 0.53 | 0.08 | −0.76** | 0.00 |
|  |  | 0.54 | 0.08 | −0.79 | 0.00 |
| Recall | r1 | 0.79 | 0.12 | 0.41* | 0.00 |
|  | r2 | 0.82 | 0.09 | 0.25 | 0.00 |
|  |  | 0.81 | 0.11 | 0.33 | 0.00 |
| F1 score | f1 | 0.63 | 0.06 | −0.74** | 0.00 |
|  | f2 | 0.64 | 0.06 | −0.85** | 0.00 |
|  |  | 0.63 | 0.06 | −0.80 | 0.00 |
| Specificity | s1 | 0.80 | 0.11 | −0.72** | 0.00 |
|  | s2 | 0.79 | 0.11 | −0.59** | 0.00 |
|  |  | 0.80 | 0.11 | −0.66 | 0.00 |

Significant at the 0.01 level (two-tailed) with **.

Significant at the 0.05 level (two-tailed) with *.

In this table, A1, p1, r1, f1, and s1 represent the outputs from the first experiment, while a2, p2, r2, f2, and s2 represent the outputs from the second experiment. Additionally, the arithmetic means of the outputs from both experiments are calculated.

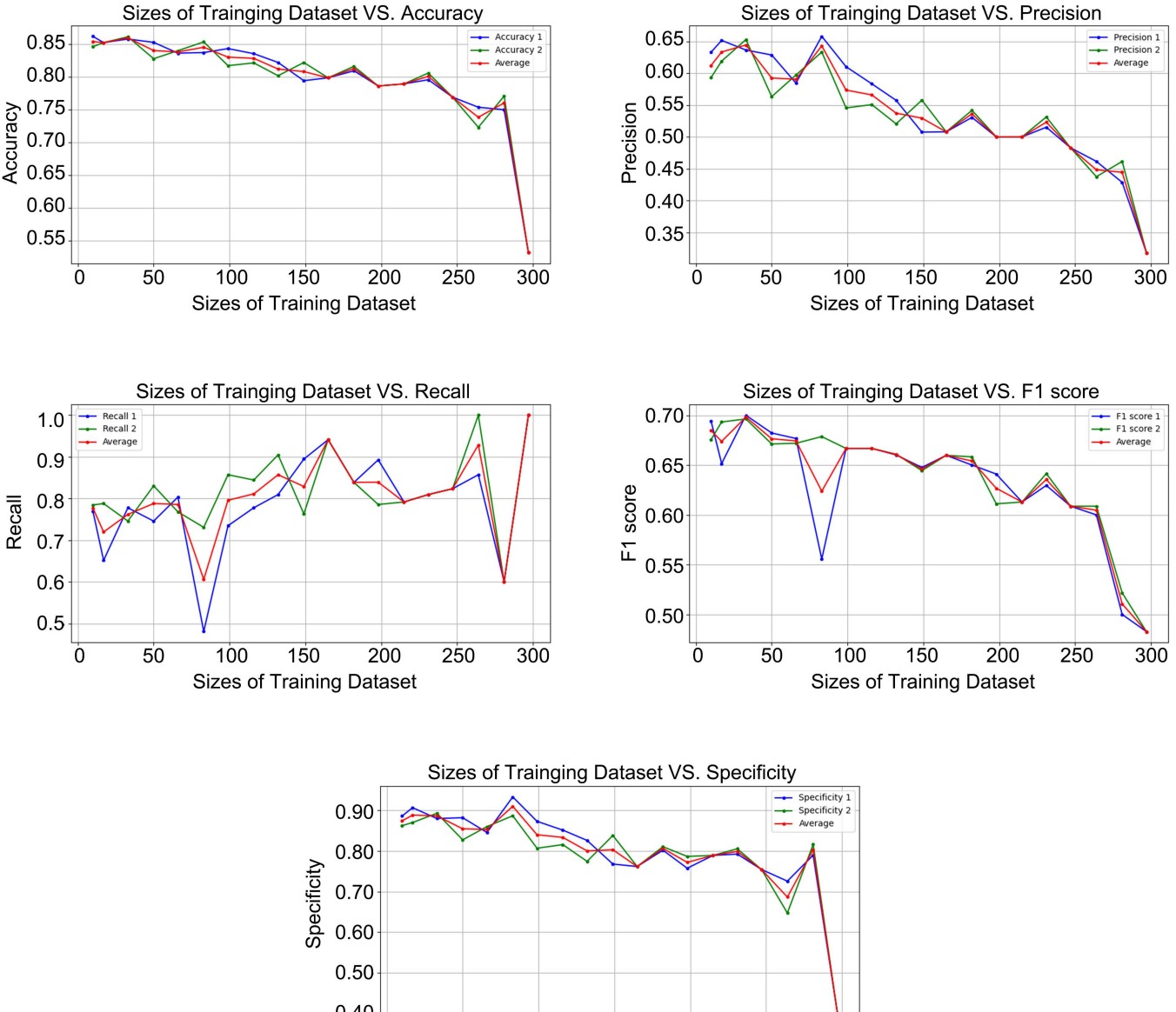

**Fig 6. Model Output Parameter Curves at Varying Training Set Sizes Fig.5 presents the outcome parameters from two distinct experimental replications, along with their computed averages.** The parameters labeled as "Accuracy 1" and "Accuracy 2" correspond to the measures of accuracy obtained from the first and second trials, respectively. The "Average" denotes the mean value of these two accuracy measures, providing an aggregate assessment of the model's performance across both trials. Similar nomenclature applies to other parameters depicted in the figure, with each set of parameters from the individual trials juxtaposed against their average to elucidate trends and consistency in the experimental outcomes. This visualization aids in evaluating the reliability and reproducibility of the model's predictive capabilities. The results and average values from the two experiments indicate that the learning curves for accuracy and specificity are relatively concentrated. In contrast, the curves for precision, recall, and F1 score each display outlier points.

## Comparative analysis of human and machine learning scoring

In this segment of the experiment, we assigned manual scores to the relevance of articles to emulate the expectations in the actual search process, using a scoring system that ranged from fully relevant to partially relevant. The Spearman correlation analysis revealed a positive correlation with human expectations (Table 6). When the size of the training set exceeded 10% of the total dataset, the p-value dropped below 0.01 for the first time, indicating a statistically significant correlation. This threshold marks a pivotal point in the performance of the machine learning model, signifying the model's enhanced ability to predict outcomes in alignment with human expectations as more data is incorporated into the training set.

At a training set proportion of 55%, the Spearman correlation coefficient reached 0.683, nearly 0.7, indicating a robust correlation. The sequence and numbering of the articles did not interfere with the outcome, suggesting that the results generated by ASReview were closely aligned with human retrieval expectations. It should be noted that in practical applications, due to the prerequisite of pre-marking the training set recognized by ASReview, obtaining a training set that represents 55% of the total is quite challenging. After all, the purpose of using the model is to eliminate a large volume of irrelevant literature and obtain the target content; there is no need to expend considerable time manually reading and labeling documents that constitute more than half of the total. In fact, once the training set exceeds 10% of the total data, a significant positive correlation between human expectations and machine output is evident. Hence, we can deduce that model outcomes, when the training set size is between 10% and 55% of the entire dataset, are likely to satisfy user expectations.

**Table 6. Spearman correlation analysis results between machine learning and human expectations.**

| Dataset | ρ | Sig. |
| --- | --- | --- |
| Default | 0.220 | 0.067 |
| 5% | 0.232 | 0.054 |
| 10% | 0.321** | 0.007 |
| 15% | 0.238* | 0.047 |
| 20% | 0.361** | 0.002 |
| 25% | 0.364** | 0.002 |
| 30% | 0.317** | 0.008 |
| 35% | 0.411** | 0.000 |
| 40% | 0.478** | 0.000 |
| 45% | 0.558** | 0.000 |
| 50% | 0.639** | 0.000 |
| 55% | 0.683** | 0.000 |
| 60% | 0.722** | 0.000 |
| 65% | 0.686** | 0.000 |
| 70% | 0.781** | 0.000 |
| 75% | 0.898** | 0.000 |
| 80% | 0.924** | 0.000 |
| 85% | 0.975** | 0.000 |
| 90% | 0.995** | 0.000 |

Significant at the 0.01 level (two-tailed) with **.

Significant at the 0.05 level (two-tailed) with *.

"Default" represents the default settings of ASReview as a control.

## External validation

For the external validation set, we retrieved entirely new data using the "most recent" strategy with the keywords "deep learning and brain injury" and "deep learning and head injury". The training set covers from January 1, 2014 to March 5, 2024, while the external validation set includes all literature from March 6, 2024 to May 25, 2025, totaling 116 articles. After data cleaning, 109 valid articles were obtained.

When we input these 109 articles into the model for processing, three experts assessed them. The experts categorized the 109 articles into 22 relevant and 87 irrelevant ones, with a Fleiss' kappa value of 0.89. Comparing the model's output with the experts' ratings, we obtained an accuracy of 0.93, precision of 0.73, recall of 1.00, F1 score of 0.85, and specificity of 0.91. These results align with those of the test set, indicating the model's stable performance and its ability to accurately identify relevant literature and exclude irrelevant ones.

## Discussion

In prior research, machine learning frameworks and other automated screening methods have predominantly been applied to clinical medicine and biology, with scant application observed in the field of forensic medicine. Compared to fields such as clinical medicine, forensic medicine is a more niche area with a relatively smaller pool of professionals and researchers, making academic output challenging to quantify. Taking the domain of "the application of deep learning in the study of injury mechanisms of cranial trauma" as an example, research advancements in clinical and biomedical fields are not readily adaptable. However, forensic medicine researchers can draw inspiration from clinical research findings and progress. This necessitates the extraction of relevant content from a vast array of unrelated literature, requiring the processing of imbalanced literature data (relevant-irrelevant), a process that is both laborious and time-consuming.

In the present study, we demonstrated that the open-source machine learning framework ASReview can be used in literature reviews of forensic medicine. It can overcome challenges in handling extremely imbalanced data. From stratified five-fold cross-validation results, ASReview demonstrates sufficient ability to automatically review and identify relevant articles on forensic brain injury modes. In other words, the model has practical value in the field of forensic medicine.

Confusion matrices support ASReview's superior capability in correctly identifying negative samples. Machine learning frameworks can help filter out completely unrelated material, such as articles from unrelated fields. After excluding a substantial amount of irrelevant content, the workload for manual review by users has been significantly reduced.

To verify the efficiency of the model in handling extremely imbalanced data and to validate the identification ability across different sizes of training datasets, we analyzed the results of two experiments and calculated their average values in the section "Assessing Model Robustness in Unbalanced Data Scenarios." The results show the model to be stable and robust in terms of accuracy and specificity. However, unstable precision and recall may lead to various irrelevant outputs, which could decrease efficiency. Therefore, the exclusion of negative samples is most significant in this experiment because it has a general correlation with the size of the training set and is less susceptible to the influence of other parameters. This observation is also consistent with the output results of the confusion matrix. We have confirmed a strong negative correlation between the size of the training set and the model's output performance. Overfitting is observed when the training set size exceeds 264 samples, which constitutes over 80% of the total dataset. Therefore, it can be concluded that when the training set size is less than 80% of the total, the specificity metric is effective, indicating that ASReview has the capability to exclude irrelevant literature.

We proceeded to conduct a comparative analysis of human and machine learning scoring to assess the divergence between machine learning outcomes and human expectations [11,20]. The results indicated that when the training set size exceeds 10% of the total data, a significant positive correlation emerges between human expectations and machine-generated outputs. Within the range of 10% to 55% of the entire dataset for training set size, the model's results meet

the expectations of users. The papers with strong relevance (Levels 1, 2, and 3), including but not limited to Paper 4 [21], Paper 7 [22], and Paper 21 [23], as well as a subset of weakly relevant papers (Level 4 and 5), have been accurately identified and outputted. The order and numbering of the papers did not affect the ASReview outcome. It is noteworthy that, taking the strongly relevant papers as examples, Paper 4 discusses the automatic detection of intracerebral hemorrhage in post-mortem CT data, whereas Paper 7 focuses on predicting skull fractures in traumatic events using finite element models. Although these papers pertain to different research directions, they are highly relevant to the search topic of this literature review. Human experts can incorporate such cases during the review process, and the machine's ability to learn from human intent and correctly output such content demonstrates that the learning efficiency and performance of ASReview have referential value and robustness. The outputs from ASReview are fundamentally in accordance with the research requirements of the users. Additionally, in response to the "user-friendly design" proposed by the original developers of ASReview, we conducted a validation by having experimenters without a background in computer science operate the model throughout the process. The users generally found the workflow easy to understand and the operations highly convenient.

We deem it essential to highlight the following issues that warrant attention in our research context:

1. The reason why negative results are considered more valuable in this experiment is noteworthy. ASReview adheres to a binary classification system, classifying each paper as either relevant or not relevant. This aligns with the logic of real-world usage and matches researchers' aims and goals for literature review. This is because there are often articles that we cannot easily determine as relevant, or some that, while not highly relevant, may provide useful content or methods. Although there are still difficulties in screening such papers through the model, using ASReview can quickly filter out irrelevant papers, significantly improving efficiency.

2. It is worth noting that the model's classification of each paper is independent, initially learning from the training set and subsequently analyzing the test set. In contrast, the thought process of human experts is continuous, and they may even be influenced by individual papers, thereby altering the ranking of other literature. Therefore, it is necessary to repeatedly train the model on target literature and adjust parameters to form a machine learning model suitable for the field of forensic science, in order to achieve higher retrieval accuracy. In the experiment, we sorted and encoded all literature within Truth 1 and Truth 2, ensuring consistent encoding throughout the training set extraction process. This consistency guarantees that ASReview's subsequent training instances include the content learned from previous iterations. Employing this incremental learning approach is a preliminary attempt to enhance the model's precision.

3. There are several limitations in this experiment that warrant attention. For instance, the capability of ASReview to learn from literature in languages other than the one used in this study has not been assessed. Furthermore, there were instances where all three experts found it challenging to rate certain papers [24]. Comparing such ratings with machine learning outcomes may not be entirely accurate.

Building on the current study, future research will further expand and deepen the evaluation of ASReview's performance from multiple perspectives. First, we aim to increase the sample size to enhance the statistical power and generalizability of our findings. In addition to parameters related to computer terminology, we will measure a broader range of system performance indicators, such as computational time, CPU load, and memory usage, to comprehensively assess the efficiency and resource consumption of ASReview in practical applications. Moreover, we will explore the development of multilingual plugins or patches for ASReview to enable text recognition and retrieval in multiple languages, thereby expanding its potential for cross-language literature searches.

Through these expansions in research directions, we hope to provide deeper insights into the development and optimization of systematic review tools and to promote their application in a wider range of academic and practical fields.

## Conclusion

In conclusion, the open-source machine learning framework ASReview has proven to be viable for the analysis of literature within the field of forensic medicine. It has the capability to accurately exclude a substantial amount of irrelevant literature, thereby enhancing the efficiency of literature review and selection processes. The platform exhibits stable performance when the proportion of labeled training set data is below 80% of the total sample size. Moreover, when this proportion ranges from 10% to 55%, the outcomes generated by ASReview align with the expectations of human researchers conducting literature searches.

## Author contributions

**Conceptualization:** Ya-Wen Liu, Zhi-Ling Tian, Ning-Guo Liu.

**Data curation:** Ya-Wen Liu, Dong-Hua Zou, He-Wen Dong, Zhi-Ling Tian, Ning-Guo Liu.

**Formal analysis:** Dong-Hua Zou, He-Wen Dong.

**Investigation:** Yuan-Yuan Liu, En-Hao Fu.

**Methodology:** Ya-Wen Liu, Zhi-Ling Tian, Ning-Guo Liu.

**Software:** Ya-Wen Liu.

**Supervision:** Dong-Hua Zou, He-Wen Dong.

**Writing – original draft:** Ya-Wen Liu.

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
