## [Decision Letter · Decision Letter 0]

Dear Dr. Liu,

Thank you for submitting your manuscript to PLOS ONE. After careful consideration, we feel that it has merit but does not fully meet PLOS ONE’s publication criteria as it currently stands. Therefore, we invite you to submit a revised version of the manuscript that addresses the points raised during the review process.

We look forward to receiving your revised manuscript.

Kind regards,

Alessandro Bruno, Ph.D.

Academic Editor

PLOS ONE

Journal requirements: When submitting your revision, we need you to address these additional requirements. 1. Please ensure that your manuscript meets PLOS ONE's style requirements, including those for file naming. The PLOS ONE style templates can be found at https://journals.plos.org/plosone/s/file?id=wjVg/PLOSOne_formatting_sample_main_body.pdf and https://journals.plos.org/plosone/s/file?id=ba62/PLOSOne_formatting_sample_title_authors_affiliations.pdf. 2. Please note that PLOS ONE has specific guidelines on code sharing for submissions in which author-generated code underpins the findings in the manuscript. In these cases, we expect all author-generated code to be made available without restrictions upon publication of the work. Please review our guidelines at https://journals.plos.org/plosone/s/materials-and-software-sharing#loc-sharing-code and ensure that your code is shared in a way that follows best practice and facilitates reproducibility and reuse. 3. PLOS requires an ORCID iD for the corresponding author in Editorial Manager on papers submitted after December 6th, 2016. Please ensure that you have an ORCID iD and that it is validated in Editorial Manager. To do this, go to ‘Update my Information’ (in the upper left-hand corner of the main menu), and click on the Fetch/Validate link next to the ORCID field. This will take you to the ORCID site and allow you to create a new iD or authenticate a pre-existing iD in Editorial Manager. 4. Thank you for stating the following financial disclosure:  [The study was financially supported by grants from the Central Research Institute Public Project (GY2024D-1, GY2024Z-1), the National Natural Science Foundation of China (82171872), Shanghai Yangfan Special Programme (23YF1448700), Shanghai Key Laboratory of Forensic Medicine (21DZ2270800), Shanghai Forensic Service Platform, Key Laboratory of Forensic Science, Ministry of Justice, the Project of Shanghai Association of Forensic Science (SHSFJD2023-008), the 2023 Open Foundation of Key Laboratory of Forensic Pathology, Ministry of Public Security, P. R. China (GAFYBL202308).].  Please state what role the funders took in the study.  If the funders had no role, please state: ""The funders had no role in study design, data collection and analysis, decision to publish, or preparation of the manuscript."" If this statement is not correct you must amend it as needed. Please include this amended Role of Funder statement in your cover letter; we will change the online submission form on your behalf. 5. In the online submission form, you indicated that [The data generated and analyzed during the study are available from the corresponding authors on reasonable request.]. All PLOS journals now require all data underlying the findings described in their manuscript to be freely available to other researchers, either 1. In a public repository, 2. Within the manuscript itself, or 3. Uploaded as supplementary information.This policy applies to all data except where public deposition would breach compliance with the protocol approved by your research ethics board. If your data cannot be made publicly available for ethical or legal reasons (e.g., public availability would compromise patient privacy), please explain your reasons on resubmission and your exemption request will be escalated for approval. 

Additional Editor Comments:

Dear Authors,

Your paper is certainly of interest and shows scientific soundness.

However, there are several major drawbacks you should work on to improve the current manuscript version.

Please run a dry English grammar check and address all remarks as in the reviewer's report.

Kind regards,

A.B.

Reviewers' comments:

**Comments to the Author**

1. Is the manuscript technically sound, and do the data support the conclusions?

Reviewer #1: Partly

2. Has the statistical analysis been performed appropriately and rigorously?

Reviewer #1: Yes

3. Have the authors made all data underlying the findings in their manuscript fully available?

Reviewer #1: Yes

4. Is the manuscript presented in an intelligible fashion and written in standard English?

Reviewer #1: Yes

Reviewer #1: A very interesting article. I hope the feedback below helps the author make this submission more impactful.

1) A sentence describing ASReview should be included in the abstract.

2) The full article needs a grammar check. Especially the tasks/methods already performed or the conclusions drawn should be in the past tense. There are several instances where there is inconsistent use of present and past tense. I recommend rectifying this for better flow and good writing.

3) Spacing after '.' at the end of sentences is inconsistent. There are also a few occurrences of additional spaces between words. Lastly, references are sometimes included after '.' and sometimes before. Please check the whole article for consistent formatting. Was this an issue with how the submission by the journal was formatted or in the original draft from the authors?

4) The introduction section can be shortened to make it concise.

5) The section "Data and models" should have model instead of models in the name as only one model is discussed in this article. Can there be a better worded heading instead of "Data and model."

6) Data collection and filtering -

a. Check the word pertaining.

b. Zotero can include a reference, the version number, and the company name. This should be sufficient and the small section "Zotero 6.0.37" under "Deep learning model and settings" can be removed.

7) Gold Standard -

a. It is a little misleading to call it dataset 1 and dataset 2 as they use the same data and just different truths (gold standard). Either change it to truth 1 and truth 2 or have an explicit sentence under this section clarifying that the data was the same and the truth was different.

b. Three senior forensic experts were used to prepare the truths. Did they have craniocerebral injury and deep learning knowledge to handle this interdisciplinary review? Are these three experts only the "specialists" mentioned in the article? If yes, either use experts or specialists throughout.

c. Did the same three specialists categorize the 70 relevant papers into the five levels?

8) Is it van de Schoot or Van De Schoot? Both are used in the article and since it is a proper noun, only one can be correct. Please correct this.

9) Deep learning model and settings -

a. Shouldn't learning have capitalized L in the heading "Active learning for Systematic Reviews toolbox" heading?

b. "The selection of these default values is justified by their consistently high performance across several benchmark experiments on various datasets." - Please add references to the benchmark experiments for this sentence showing the use of these default values.

c. It is also mentioned that the default settings have short computation time - recommend including the actual time taken on average - can also be a range of time taken.

10) Methods - Stratified 5-fold cross-validation

a. Reword the last paragraph in this paragraph as the sentences seem redundant.

b. Accuracy is not an ideal measure as the dataset is imbalanced. Why was AUC from the ROC curve not used as a performance metric given that it is not dependent on prevalence? Recommend reporting the AUC metric for all analyses reported in this article.

11) Does "encoding order" actually mean the sequential numbering of the 329 documents mentioned in the "Data collection and filtering" section? If yes, please specify this clearly under the data collection section rather than using it directly at the end of Methods. Also, can another phrase such as indexing or similar be used since "encoder" in AI means different?

12) Is there other literature showing the use of the "snowballing" approach? If yes, please reference them with "snowballing" in the article.

13) A minor suggestion is to avoid using words such as demystify/admirably and keep them more technical or appropriate for scientific writing.

14) Comparative Analysis of Human and Machine Learning Scoring

a. Please clarify in the article that when the selection proportion was increased by 5% randomly for "Comparative Analysis of Human and Machine Learning Scoring," it was ensured that the previously selected data was retained and not selected again newly in the following iterations.

b. Also clarify ground truth for this - explaining that levels 1 to 3 of the experts were taken as relevant and levels 4 and 5 were taken as irrelevant.

15) "model's superior capability" under the Stratified 5-fold cross-validation section of Results should avoid using the word superior since there was no statistical test performed to show its superiority during 5-fold.

16) Were the p-values adjusted for multiple comparisons? If yes, please specify and if no, justify why not.

17) "The Spearman correlation analysis revealed a positive correlation between human expectations (Table 6)." It will not be between human expectations; it will be with human expectations. Please check for grammar throughout.

18) Suggestion to include future scope in the discussion section.

**Do you want your identity to be public for this peer review?** For information about this choice, including consent withdrawal, please see our Privacy Policy

Reviewer #1: No

---

## [Author Response · Author response to Decision Letter 1]

18 Mar 2025

Thank you very much for your attention to our manuscript and your valuable comments. Please read the file"Response to reviewers".

---

## [Decision Letter · Decision Letter 1]

Dear Dr. Liu,

Thank you for submitting your manuscript to PLOS ONE. After careful consideration, we feel that it has merit but does not fully meet PLOS ONE’s publication criteria as it currently stands. Therefore, we invite you to submit a revised version of the manuscript that addresses the points raised during the review process.

**ACADEMIC EDITOR: **

address all comments remarked by reviewerspoint out how you chose the state-of-the-art references in your manuscriptstate what you mean by "reasonable request" for data accessibility

We look forward to receiving your revised manuscript.

Kind regards,

Alessandro Bruno, Ph.D.

Academic Editor

PLOS ONE

Additional Editor Comments:

Dear Authors,

The new manuscript version is improved. Thanks for your work.

However, there are some points remarked by two reviewers that need your attention.

Please look at both reviewers' report and deal with them.

Kindest regards,

A.B.

Reviewers' comments:

Reviewer's Responses to Questions

**Comments to the Author**

Reviewer #1: All comments have been addressed

Reviewer #2: All comments have been addressed

Reviewer #3: (No Response)

2. Is the manuscript technically sound, and do the data support the conclusions?

Reviewer #1: Yes

Reviewer #2: Yes

Reviewer #3: No

3. Has the statistical analysis been performed appropriately and rigorously?

Reviewer #1: Yes

Reviewer #2: Yes

Reviewer #3: I Don't Know

4. Have the authors made all data underlying the findings in their manuscript fully available?

Reviewer #1: Yes

Reviewer #2: Yes

Reviewer #3: No

5. Is the manuscript presented in an intelligible fashion and written in standard English?

Reviewer #1: Yes

Reviewer #2: No

Reviewer #3: Yes

Reviewer #1: All comments have been addressed by the authors appropriately. The paper can be accepted for publication as submitted.

Reviewer #2: This manuscript presents an intriguing study that evaluates the performance of the machine learning tool ASReview for literature screening in the field of forensic science. Particularly noteworthy is the demonstration that artificial intelligence, specifically deep learning, can function effectively even in the complex and often scientifically ambiguous domain of forensic medicine. The observation that overfitting—akin to a human-like state of "overstudying" or absorbing excessive information—may actually degrade performance adds a novel and socially relevant dimension to the work, enhancing its originality and significance.

However, the abstract as currently written is somewhat difficult to follow, largely due to its use of technical jargon and abstract structure. This could obscure the central message of the study, especially for readers without a strong background in either medicine or data science. Therefore, I strongly recommend revising the abstract in a more accessible and structured manner, such as the following example:

Suggested Abstract (Structure and Example)

"In recent years, the explosive growth of academic publications has made it increasingly important to efficiently extract relevant information across all fields. This task is particularly challenging in forensic medicine, a discipline that requires interdisciplinary knowledge and frequently deals with scientifically inconclusive cases—placing a heavy burden on literature reviewers.

In response to this issue, the present study investigates whether ASReview, a machine learning-based automated literature screening tool, can be effectively applied to forensic science. As a simulated topic, we selected 'the application of deep learning in traumatic brain injury research' and analyzed 329 publications using stratified five-fold cross-validation.

The results showed that ASReview performed well, especially in automatically excluding irrelevant literature. When the proportion of labeled training data ranged from 10% to 55% of the entire dataset, the model’s output closely aligned with expert human judgment. However, when the proportion exceeded 80%, overfitting occurred, resulting in a decline in performance. This phenomenon—where the AI 'overlearns' and becomes less accurate—echoes real-world cognitive patterns in human learning and highlights both the promise and the limits of AI-assisted screening.

Overall, ASReview demonstrated practical utility even with limited training data, and it has the potential to significantly reduce the workload of researchers in forensic medicine—a field marked by complex and diverse literature."

A revision following this kind of structure—clearly stating the background, purpose, methods, results, and implications—would help make the abstract more compelling and understandable for a broader audience. I also suggest simplifying the overall writing style and minimizing the use of highly technical terminology where possible.

The study makes a valuable and empirical contribution by demonstrating the feasibility of AI-assisted literature review in the field of forensic science. I recommend acceptance pending revision, particularly with regard to improving the readability of the abstract and presentation of results.

Additionally, I strongly encourage the authors to revise the figure summarizing the five-tier relevance ranking to be more intuitive and immediately interpretable at a glance.

Reviewer #3: Writing:

1. There are numerous minor typos, missing characters and grammatical errors throughout manuscript.

Methods:

In general, there seems to be a lack of rigor (blinding/masking, randomization, exclusion criteria, data availability etc) in the study design.

1. How were search results sorted in Pubmed (best match? most recent? author name? etc).

2. Was search and retrieval automated or manual? If manual, were personnel blinded to author names, titles, journals etc. In what order were papers imported into Zotero?

3. What additional steps, if any, were taken to avoid bias in the article listing order in Zotero?

4. How are "problematic papers" defined? The criteria should be clearly listed. Were these criteria determined prior to conducting the literature search?

5. Were ratings by three experts averaged or was there a tie-breaker?

6. Was inter-rater reliability calculated for the three expert reviewers?

7. Was dataset 1 shuffled randomly prior to splitting into 5 subsets for 5-fold CV?

8. Preparation of data (removal of "problematic papers", using Zotero's filtering capabilities etc) should be performed on the assigned training subset within each CV loop, rather than on the broader dataset1, to avoid data leakage.

9. The trained model should be prospectively assessed in a completely novel dataset that has not been previously curated by human experts. The resulting article classifications should then be reviewed by the human experts post-hoc to determine the model's real-world predictive performance.

Data Availability:

1. Data is not fully available without restriction. Authors state the data are "available upon reasonable request". Who decides what a reasonable request is? This stance does not align with the authors' answer in the Data Availability Statement, and is contrary to PLOS Data policy.

**Do you want your identity to be public for this peer review?** For information about this choice, including consent withdrawal, please see our Privacy Policy

Reviewer #1: No

Reviewer #2: No

Reviewer #3: No

---

## [Author Response · Author response to Decision Letter 2]

4 Jun 2025

Thank you for your review.

Your insightful suggestions have provided us with valuable guidance.

Please refer to the document "response to reviewers" for the detailed responses to the reviewers' comments.

---

## [Decision Letter · Decision Letter 2]

Dear Dr. Liu,

Thank you for submitting your manuscript to PLOS ONE. After careful consideration, we feel that it has merit but does not fully meet PLOS ONE’s publication criteria as it currently stands. Therefore, we invite you to submit a revised version of the manuscript that addresses the points raised during the review process.

Address the latest minor issues left in the manuscript.

plosone@plos.org

We look forward to receiving your revised manuscript.

Kind regards,

Alessandro Bruno, Ph.D.

Academic Editor

PLOS ONE

Journal Requirements:

**Additional Editor Comments:**

Dear authors,

I invite you to address the minor issues raised by the reviewers.

I recommend your manuscript for a minor revision round.

Kind regards,

A.B.

Reviewers' comments:

Reviewer's Responses to Questions

**Comments to the Author**

Reviewer #2: All comments have been addressed

Reviewer #4: All comments have been addressed

2. Is the manuscript technically sound, and do the data support the conclusions?

Reviewer #2: Yes

Reviewer #4: Yes

3. Has the statistical analysis been performed appropriately and rigorously?

Reviewer #2: Yes

Reviewer #4: Yes

4. Have the authors made all data underlying the findings in their manuscript fully available?

Reviewer #2: Yes

Reviewer #4: No

5. Is the manuscript presented in an intelligible fashion and written in standard English?

Reviewer #2: Yes

Reviewer #4: Yes

Reviewer #2: All comments have been addressed by the authors appropriately.

The paper can be accepted for publication as submitted.

Reviewer #4: the manuscript is technically sound, and the presented data adequately support its conclusions—with a few caveats and suggestions for improvement.

The statistical analysis appears to be appropriate and rigorous

I can’t confirm whether the data are fully available. If they’re not deposited in a public repository and documented in the manuscript, they haven’t met strong open-data standards.

The manuscript is clearly organized and readable

**Do you want your identity to be public for this peer review?** For information about this choice, including consent withdrawal, please see our Privacy Policy

Reviewer #2: **Yes: ** Keiichi Abe

Reviewer #4: **Yes: ** Khalid Waleed Abdo

---

## [Author Response · Author response to Decision Letter 3]

10 Jul 2025

Thank you for your careful review and valuable comments regarding our manuscript.

Your insightful suggestions have provided us with valuable guidance.

Please read the document titled "Response to reviewers".

---

## [Editor Report · Decision Letter 3]

An Open-Source Interactive AI Framework for Assisting Automatic Literature Review in Forensic Medicine: Focus on Brain Injury Mechanisms

PONE-D-24-60438R3

Dear Dr. Liu,

We’re pleased to inform you that your manuscript has been judged scientifically suitable for publication and will be formally accepted for publication once it meets all outstanding technical requirements.

Kind regards,

Alessandro Bruno, Ph.D.

Academic Editor

PLOS ONE

Additional Editor Comments (optional):

Dear Authors,

Thanks for fixing all the minor issues raised by the reviewers.

I will recommend your paper for acceptance.

Kindest regards,

A.B.
---

## [Editor Report · Acceptance letter]

PONE-D-24-60438R3

PLOS ONE

Dear Dr. Liu,

I'm pleased to inform you that your manuscript has been deemed suitable for publication in PLOS ONE. Congratulations! Your manuscript is now being handed over to our production team.

Kind regards,

on behalf of

Associate Professor Alessandro Bruno

Academic Editor

PLOS ONE